# Optimization of experimental designs for biological rhythm discovery

Turner Silverthorne[1,2,3], Matthew Carlucci[2,3], Arturas Petronis[2,3], Adam R. Stinchcombe[1]*

1 Department of Mathematics, University of Toronto, Toronto, Ontario, Canada, 2 The Krembil Family Epigenetics Laboratory, The Campbell Family Mental Health Research Institute, Centre for Addiction and Mental Health, Toronto, Ontario, Canada, 3 Institute of Biotechnology, Life Sciences Center, Vilnius University, Vilnius, Lithuania

* stinch@math.toronto.edu

## Abstract

Equally spaced temporal sampling is the standard protocol for the study of biological rhythms. These equispaced designs perform well when calibrated to an oscillator's period, yet can introduce systematic biases when applied to rhythms of unknown periodicity. Here, we investigate how optimizing the timing of measurements can improve rhythm detection across a range of experimental settings. When the period of a rhythm is known, we prove that equispaced designs provide optimal statistical power. In studies targeting specific sets of candidate rhythms, we construct optimal alternatives to equispaced designs to simultaneously maximize power at all frequencies under consideration. For studies investigating continuous period ranges, we show numerically how blindspots near the Nyquist rate can be resolved through timing optimization. Our computational methods are available through our Power-CHORD library. Our findings across all experimental contexts suggest that timing optimization is an effective yet under-explored tool for improving biological rhythm discovery.

## Author summary

Biological systems often exhibit fluctuations when observed over time. Statistical tests can help to determine whether these fluctuations are evidence of an underlying biological cycle or attributable to noise. The sensitivity of these tests depends not only on the sheer number of observations, but also on when observations are taken along the cycle. We confirm that the standard practice of making observations at equal intervals along the cycle is indeed the most sensitive design for a fixed sample size, however, this approach is only tenable for cycles of known duration. When we attempted to extend standard practices to the context of discovering cycles of unknown length, we uncovered significant drawbacks that

**Data availability statement:** All relevant data are within the manuscript and its Supporting Information files. Implementations of PowerCHORD in R and MATLAB are available through the first author's GitHub, https://github.com/t-silverthorne/PowerCHORD.

**Funding:** TS is supported by an NSERC Canada Graduate Scholarship (www.nserc-crsng.gc.ca). This work was supported by the Krembil Foundation, Toronto, Canada (krembilfoundation.ca), the Future Biomedicine Charity Fund, Vilnius, Lithuania (abfondas.lt/en/), and by Research Council of Lithuania (lmt.lrv.lt/en) under the Programme "University Excellence Initiatives" of the Ministry of Education, Science and Sports of the Republic of Lithuania (No. 12-001-01-01-01 "Improving the Research and Study Environment"; project No S-A-UEI-23-10). AP is a Marius Jakulis Jason Foundation scholar (mjjfondas.lt/en). ARS acknowledges the support of the Natural Sciences and Engineering Research Council of Canada (NSERC): RGPIN-2019-06946 (www.nserc-crsng.gc.ca). This research was enabled in part by support provided by Compute Ontario and the Digital Research Alliance of Canada (www.alliancecan.ca). None of these funders played any role in the study design, data collection and analysis, decision to publish, or preparation of the manuscript.

**Competing interests:** The authors have declared that no competing interests exist.

would lead to meaningful signals being overlooked. We overcame these limitations of equispaced measurements by developing a mathematical optimization framework that is applicable when cycle length is unknown or when equispaced designs are infeasible. Solving this problem numerically for a range of experimental conditions produced designs that have the potential to expedite the discovery of novel biological rhythms.

## 1 Introduction

Biological rhythms serve essential roles in living systems and arise from diverse mechanisms on scales ranging from individual cells to entire populations [1–4]. While familiar examples such as circadian rhythms, somitogenesis, and the cell-cycle have been studied for decades [5–7], new rhythms and layers of temporal organization continue to be discovered [8–12]. In the context of circadian studies, rhythm discovery has been facilitated in part by improvements to statistical methods [13–16], leading to evidence of new rhythmicity in existing datasets [17]. These improvements are now a major focus of rhythm discovery guidelines [18,19], however the dependence of statistical performance on the underlying experimental design is less thoroughly explored. Hence, many studies make use of equispaced temporal sampling as the *de facto* design choice.

The preferred status of equispaced designs is justifiable by the strong assumptions of circadian studies. In particular, if the period of an oscillator is known ahead of time, designs with measurements equispaced along the oscillator's cycle achieve statistically optimal performance [20]. Yet, in more general contexts, biological oscillations are known to occur with periods ranging from milliseconds to years and a study may need to consider potential cycles across these vast timescales. Since equispaced collection along only one cycle may be unreliable for rhythm detection due to statistical power variability at nominal periods and acrophases (S1 Fig), there is a need for methods that enable optimal detection of cycles with unknown periodicity.

In this article, we address two related research objectives. The first is to develop a rigorous understanding of the experimental conditions under which equispaced designs provide optimal power for rhythm detection. The second is to provide numerical methods for constructing optimal or near-optimal designs in experimental contexts where equispaced designs fail to achieve optimal power. We present a collection of optimization method for addressing this latter objective. Our methods are available through our open-source PowerCHORD (*P*ower analysis and *C*osinor design optimization for *HO*moscedastic *R*hythm *D*etection) repository and can be applied to three types of rhythm detection experiments:

1. **Known-period:** For a rhythm detection experiment investigating a single period and aiming to achieve the highest possible power across all acrophases, we prove in Sect 3.1 that equispaced designs achieve optimal power under the assumptions of the cosinor model. We construct optimal alternatives to demonstrate how power can be balanced with experimental constraints that prohibit equispaced collection. For non-sinusoidal rhythms, we show numerically that equispaced designs continue to outperform irregular designs provided that the study has no prior information about the acrophase of the signal.

2. **Discrete-period uncertainty:** The experiment investigates a predetermined list of periods, such as harmonics of a known rhythm or a collection of environmental rhythms (circadian, circalunar, and circannual). We optimize such studies in Sect 3.2 by deriving a mixed-integer conic program equivalent to power optimization. Numerical solutions to the conic program reveal that certain groups of periods can be measured simultaneously without trade-offs in power.

3. **Continuous period uncertainty:** The experiment focuses on rhythm detection across a continuous range of periods, such as hourly to circadian periods. To accommodate the broad period uncertainty, we measure power using the free-period model under permutation testing. We compare rigorous and heuristic optimization methods for maximizing permutation power. Both methods generate designs with improved power when the frequency window includes the Nyquist rate of an equispaced design with equivalent sample size.

To make our work accessible, we focus mainly on experimental applications while providing only brief summaries of the relevant theoretical work as necessary. Proofs of the theoretical results are provided in S1 Text. Readers can explore further applications to design problems using the PowerCHORD open source code repository.

## 2 Background

### 2.1 Harmonic regression

Harmonic regression is a popular statistical framework for studying systems with oscillatory features [21]. We perform harmonic regression using the fixed-period cosinor model. For a MESOR (*M*idline *E*stimating *S*tatistic of *R*hythm) $Y_0$, amplitude $A$, acrophase $\phi$, and frequency $f$, the fixed-period cosinor model takes the form

$$Y(t) = Y_0 + A\cos(2\pi ft - \phi) + \varepsilon(t), \tag{1}$$

in which $\varepsilon(t) \sim \mathcal{N}(0, \sigma)$ is homoscedastic Gaussian white noise. Assuming the frequency $f$ is fixed, Eq 1 can be rewritten as

$$Y(t) = \beta_0 + \beta_1\cos(2\pi ft) + \beta_2\sin(2\pi ft) + \varepsilon(t), \tag{2}$$

so that all unknown parameters appear linearly. Given data $\mathbf{y} = \{y_i\}_{i=1}^N$ measured at times $\mathbf{t} = \{t_i\}_{i=1}^N \subset [0,1]^N$, where $N$ is the sample size of the experiment, the optimal least squares solution $\hat{\beta}$ for estimating the coefficients $\beta = (\beta_0, \beta_1, \beta_2)$ appearing in Eq 2 is

$$\hat{\beta} = \hat{\beta}(\mathbf{t}; f) = (X^TX)^{-1}X^T\mathbf{y}, \qquad X = X(\mathbf{t}; f) = \begin{bmatrix} \mathbf{1} & \cos(2\pi f\mathbf{t}) & \sin(2\pi f\mathbf{t}) \end{bmatrix}. \tag{3}$$

We assume $\mathbf{t}$ contains measurements at sufficiently many distinct phases for Eq 3 to be well-defined (S1 Text Lemma S1-1.1). For most of our analysis, we only consider signals with frequency $f < f_{\text{Nyq}}$ where $f_{\text{Nyq}} = N/2$ is the Nyquist rate of

an equispaced design containing $N$ measurements. Estimates of the signal's amplitude and acrophase can be computed from $\hat{\beta}$ to obtain

$$\hat{A} = \sqrt{\hat{\beta}_1^2 + \hat{\beta}_2^2}, \quad \hat{\phi} = \text{atan2}\left(\hat{\beta}_2, \hat{\beta}_1\right). \tag{4}$$

## 2.2 Power analysis

Using the fixed-period cosinor model, rhythm detection can be formulated as a hypothesis test with null hypothesis $\beta_1 = 0 = \beta_2$ and alternative $\beta_1 \neq 0$ or $\beta_2 \neq 0$. The statistical power quantifies how reliably oscillations are detected by the experimental design and hypothesis test.

**Definition 2.1** (Statistical power of the fixed-period cosinor model). Given a parametric model with parameters $\beta \in \mathbb{R}^p$, data $Y \in \mathbb{R}^N$ measured at times $\mathbf{t} \in \mathbb{R}^N$, and a rejection region $R \subset \mathbb{R}^N$, the power of the hypothesis test is given by the probability of the data lying within the rejection region

$$\gamma(\mathbf{t}; \beta) = \mathbb{P}_\beta(Y \in R). \tag{5}$$

For the fixed-period-cosinor-based hypothesis test, the rejection region is $R = \{\mathbf{x} : \hat{F}(\mathbf{x}) \geq c\}$, in which $\hat{F}(\mathbf{x})$ is the F-statistic

$$\hat{F}(\mathbf{x}) = \frac{TSS - RSS}{RSS} \frac{N - p}{p - 1}, \tag{6}$$

$TSS = ||\mathbf{x} - \langle\mathbf{x}\rangle||^2$, $RSS = ||\mathbf{x} - X\hat{\beta}||^2$, $\langle\mathbf{x}\rangle = \frac{1}{N}\sum_{i=1}^N x_i$, $X = X(\mathbf{t}; f)$ is the cosinor design matrix and $\hat{\beta}$ is the least-squares estimate of the parameters from Eq 3.

Since the exact parameters of the signal are rarely well-known when the design is constructed, we seek designs that achieve high power across a range of parameter values. To this end, we quantify performance using the worst-case power of the design, meaning the lowest power across all signals of interest. Prioritization of worst-case rather than average power ensures that the power is above a known threshold for all relevant signals. A formal definition of worst-case power is given below. For the remainder of the paper we will simply use the terminology "optimal power designs" to refer to optimality with respect to a worst-case scenario.

**Definition 2.2** (Optimal worst-case power). Given a domain $\mathcal{B} \subset \mathbb{R}^p$ in parameter space, a design matrix $X \in \mathbb{R}^{N \times p}$, and a power function $\gamma(\mathbf{t}; \beta)$, an experimental design $\mathbf{t}^*$ achieves optimal worst-case power with respect to $\mathcal{B}$ if it satisfies

$$\mathbf{t}^* = \arg\max_{\mathbf{t} \in [0,1]^N} \min_{\beta \in \mathcal{B}} \gamma(\mathbf{t}; \beta). \tag{7}$$

Our results are organized based on the degree of period uncertainty at the time of study design. In the simplest setting, the period is treated as known and the design is optimized to detect signals of various acrophases and amplitudes for the predetermined period. The latter two contexts, defined below, assume a uniform prior distribution on the candidate periods, meaning that no period is prioritized higher than the others.

**Definition 2.3** (Discrete period uncertainty). An experiment that investigates a finite list of periods $T_1, \dots, T_k$ is said to have discrete period uncertainty.

**Definition 2.4** (Continuous period uncertainty).    An experiment that investigates all periods in a given range $T_{min} \leq T \leq T_{max}$ is said to have continuous period uncertainty.

When considering continuous period uncertainty, we test for rhythmicity by applying permutation tests to the free-period cosinor model (Sect 5.1). The models of period uncertainty we consider are guided by the type of experiment we aim to optimize. We assume that the experiment is an exploratory investigation of a sparsely characterized system, with limited prior knowledge of the underlying parameters. This assumption is also relevant to our choice to prioritize the cosinor model. Several alternative approaches to the cosinor model have been introduced [13,14,22–25]. Alternative models can be particularly useful for systems that violate the assumptions of the cosinor model, for instance biological rhythms that display non-stationary peak-to-peak duration [26]. We focus on the cosinor model because it is simple, popular, and broadly applicable, making it well suited for initial studies of sparsely characterized systems. Data collected in an exploratory study could inform the optimization of a follow-up study focused on questions beyond rhythm detection.

## 3 Results

### 3.1 Rhythms of known period

We construct optimal designs for known-period experiments by maximizing a closed-form expression for the statistical power. The power expression can be evaluated much faster than Monte Carlo power estimation and generalizes an earlier formula [27] which is only valid when measurements are equispaced (S2 Fig). A related generalization of [27] presented in [28] was obtained independently of our work. The reader is directed to S1 Text for a proof of Theorem 3.1 and the subsequent theorems.

**Theorem 3.1** (Power of the one-frequency cosinor model). *Consider the one-frequency cosinor model*

$$y(t) = \beta_0 + \beta_1 \cos(2\pi f t) + \beta_2 \sin(2\pi f t) + \varepsilon(t), \quad \varepsilon \sim \mathcal{N}(0, \sigma^2) \tag{8}$$

*applied to data* $\mathbf{y} = \{y_i\}_{i=1}^N$ *collected at distinct times* $\mathbf{t} = (t_i)_{i=1}^N$ *with $N > 3$. Suppose the following hypotheses are tested using an F-test*

- *null hypothesis $H_0 : \beta_1 = 0 = \beta_2$,*
- *alternative hypothesis $H_1 : \beta_1 \neq 0$ or $\beta_2 \neq 0$.*

*Given parameters $\beta = (\beta_0, \beta_1, \beta_2)$, the power $\gamma$ of this hypothesis test is given by*

$$\gamma(\mathbf{t}; \beta, f, \sigma) = 1 - F_{\lambda(\mathbf{t};\beta,f,\sigma)}\left(F_0^{-1}(1-\alpha; 2, N-3); 2, N-3\right), \tag{9}$$

$$\lambda(\mathbf{t}; \beta, f, \sigma) = \frac{1}{\sigma^2}\beta^T H^T (H(X^T X)^{-1} H^T)^{-1} H\beta, \tag{10}$$

*in which $X = X(\mathbf{t}; f) = \begin{bmatrix} 1 & \cos(2\pi f \mathbf{t}) & \sin(2\pi f \mathbf{t}) \end{bmatrix}$ is the design matrix, $\alpha \in (0, 1)$ is the type I error rate, $F_\lambda(x; n_1, n_2)$ is the noncentral F-distribution with $(n_1, n_2)$ degrees of freedom and noncentrality parameter $\lambda$, $F_0(x; n_1, n_2) = F(x; n_1, n_2)$ is the F-distribution, and*

$$H = \begin{bmatrix} 0 & 1 & 0 \\ 0 & 0 & 1 \end{bmatrix} \tag{11}$$

*is the hypothesis matrix.*

Analysis of Eqs 9 and 10 in S1 Text Sect S1-1.3 shows that worst-case power maximization in the sense of Definition 2.2 and Theorem 3.1 is equivalent to Elfving optimality [29]. As a consequence of this equivalence, we obtain a simple condition for equispaced designs to be optimal and provide the same power at all acrophases.

**Theorem 3.2** (Optimality condition for equispaced designs). *Suppose a study aims to detect signals of a specific frequency and unknown acrophase. Maximizing the worst-case power across all acrophases is equivalent to the following eigenvalue optimization problem*

$$\mathbf{t}^* = \mathrm{argmax}_{\mathbf{t} \in [0,1]^N} \xi_{\min}\left(B(\mathbf{t}; f)^{-1}\right) \tag{12}$$

*in which $\xi_{\min}(\cdot)$ denotes the smallest eigenvalue and the matrix $B(\mathbf{t}; f)$ is given by*

$$B(\mathbf{t}; f) = H\left(X(\mathbf{t}; f)^T X(\mathbf{t}; f)\right)^{-1} H^T. \tag{13}$$

*If it is possible to collect N > 3 equispaced measurements per cycle, then equispaced designs maximize Eq 12. Moreover, such designs maintain constant power as a function of acrophase (S3A Fig), at a constant value determined by the noncentrality parameter*

$$\lambda_{\mathrm{opt}} = \frac{A^2 N}{2\sigma^2}. \tag{14}$$

It may seem strange that the formulation of power optimization in Theorem 3.2 does not require knowledge of the amplitude or noise strength. This is because the influence of measurement timing on power is essentially independent of amplitude and noise strength. To be precise, the ratio between amplitude and noise strength simply rescales the noncentrality parameter. Since the power function depends monotonically on this parameter, the rescaling therefore does not affect the location of its critical points with respect to measurement timing. While amplitude and noise strength are of course needed for determining the precise level of power, they are not needed for the identification of designs that maximize power.

While Theorems 3.1 and 3.2 are broadly applicable to cosinor rhythms, many biological rhythms do not strictly satisfy the assumptions of the cosinor model. To understand the extent to which equispaced optimality requires cosinor assumptions, we compared the performance of equispaced and random designs on simulated non-cosinor signals. We maintained the assumption that the study had no prior knowledge of the signal's acrophase and found that equispaced designs remained optimal or at least comparable to random designs (S4 Fig). The connection between uniform acrophase uncertainty and equispaced optimality is discussed further in S1 Text Sect S1-1.6, where we demonstrate that acrophase invariance implies equispaced optimality for a more general class of optimality criteria.

In practice, studies may need to consider sub-optimal designs in order to balance power with other aspects of study design. For instance, a circadian study with human participants may aim to minimize the number of times a participant is awoken during the night for sample collection while ensuring that rhythms of all acrophases can still be reliably detected. Since strong acute disruptions of sleep have been shown to impact various biological measurements [30,31], accurate characterization of naturally occurring oscillations should avoid disruptions to a 6-12hr window of a subject's regular sleep routine.

Incorporating a rest-window reduces the power relative to equispaced designs, so we aimed to determine how much power could be recovered by optimizing the measurement times under the timing constraint. To simplify the design space, we assumed that measurements could only be collected at times aligned with a half-hour grid. Consequently, it was possible to find optimal designs by running a brute-force search (Sect 5.2.1). To gauge the benefit of timing optimization, we compared the optimal designs to "naive" designs in which all measurements are equispaced outside the rest-window (Fig 1A). Relative to the naive designs, the power of the constrained-optimal designs was considerably less sensitive to

the acrophase of the signal (Fig 1B). This difference in sensitivity can be observed in the peak-to-trough power variability of the two designs ($\Delta_{\text{naive}} = 22.93\%$, $\Delta_{\text{optimal}} = 0.35\%$; 8hr rest-window, amplitude $A = 2.5$). Comparing the noncentrality parameters of the constrained-optimal and naive designs to an equispaced design allowed us to estimate how much the constraints limit the power. Negligible power was lost in the constrained-optimal design when imposing the shortest window duration. As the window length increased, both the constrained-optimal and naive designs lost power (Fig 1C). Still, the constrained-optimal designs out-performed the naive designs across the windows under consideration, suggesting that timing optimization can help studies accommodate timing constraints without unnecessarily reducing their power.

### 3.2 Discrete period uncertainty

Experiments may prioritize the detection of specific periods based on prior knowledge of the biological system. For instance, a study may investigate harmonics of a known rhythm [12] or assume that rhythms are entrained to environmental cues (i.e. circadian, circalunar, and circannual cycles [32]). To incorporate discrete period uncertainty in our design process, we score designs based on their lowest power across all acrophases and frequencies of interest, using an objective function of the form

$$J(\mathbf{t}) = \min_{f \in \mathcal{F}} \xi_{\min}\left(B(\mathbf{t}; f)^{-1}\right),\tag{15}$$

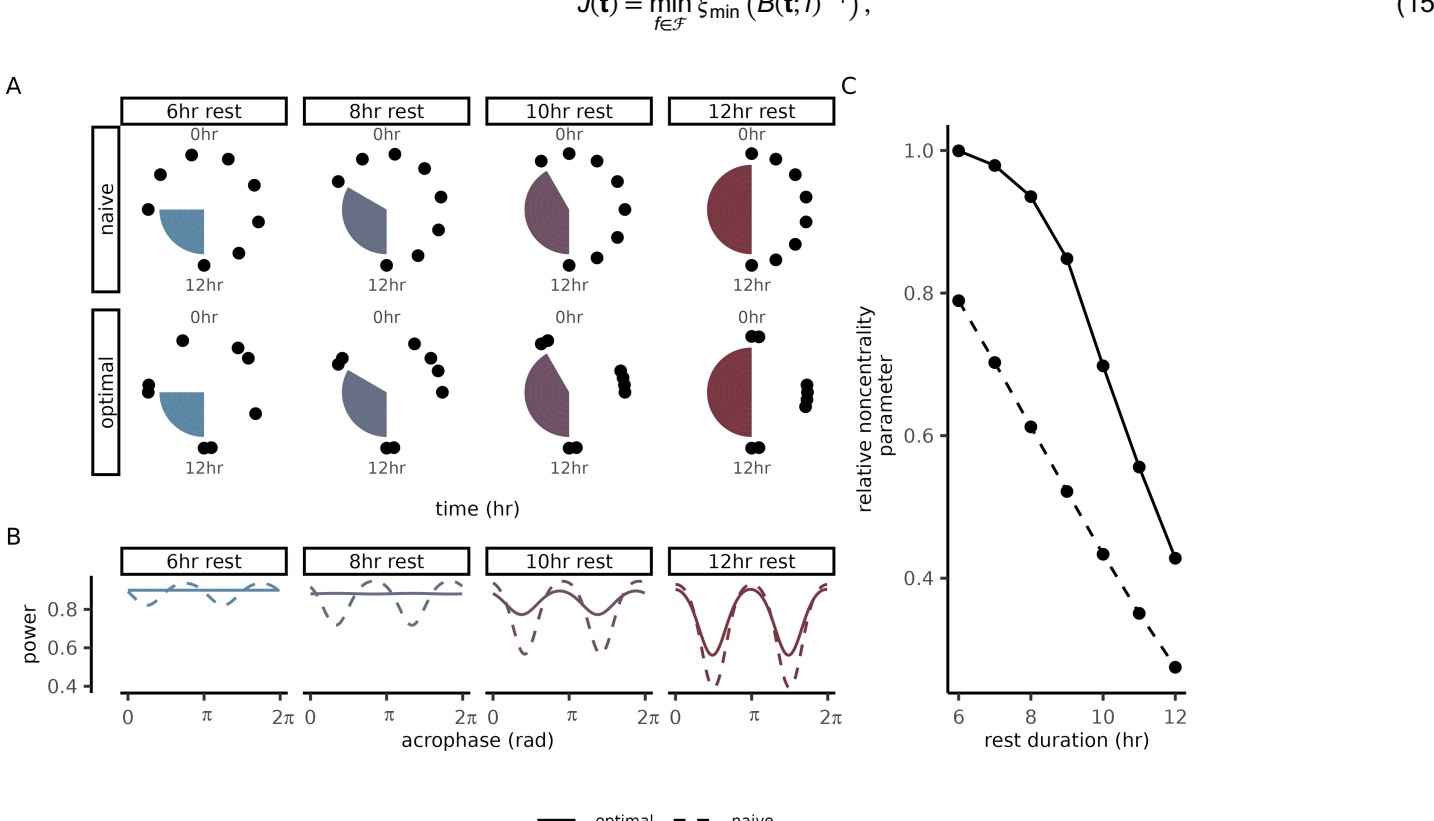

**Fig 1. Circadian studies can optimize collection times to balance power and rest-window duration. (A)** Measurement times for optimal and naive designs plotted as phases of a 24hr cycle. Each panel corresponds to a different duration of the rest-window (shaded region), during which no samples can be collected. Optimal designs were generated using a brute-force search with $N_t = 48$ points in the temporal discretization (Sect 5.2.1). Naive designs were constructed by distributing all measurements outside the rest window at equal time-intervals. **(B)** Power (y-axis) as a function of acrophase (x-axis) for each duration of the rest-window. The color of each curve represents rest-window duration and the line-type represents if the design is optimal or naive. **(C)** The noncentrality parameter (y-axis) as a function of rest-window duration (x-axis) for naive and optimal designs. Noncentrality parameters are reported relative to the noncentrality parameter of an unconstrained equispaced design with the same sample size ($N = 8$ samples).

in which $\mathcal{F} = (f_1, \ldots, f_m)$ is a collection of frequencies of interest, $B(\mathbf{t}; f)$ is the the matrix given in Eq 13, and $\xi_{\min}(\cdot)$ maps a matrix to its smallest eigenvalue. Continuing to focus on worst-case optimality (Definition 2.2), we seek designs that achieve the greatest possible value of $J(\mathbf{t})$ across all signals of interest. The connection between $J(\mathbf{t})$ and power optimization is made explicit in Corollary 3.2.1.

**Corollary 3.2.1.** *Maximizing the objective function in Eq 15 is equivalent to maximizing the worst-case power over frequencies $f \in \mathcal{F}$ and acrophase. The value achieved by this optimal solution $\mathbf{t}^* = \arg\max_{\mathbf{t} \in [0,1]^N} J(\mathbf{t})$ is bounded by*

$$J(\mathbf{t}^*) = \max_{\mathbf{t} \in [0,1]^N} \left[ \min_{f \in \mathcal{F}} \xi_{\min} \left( B(\mathbf{t}^*; f)^{-1} \right) \right] \leq \frac{A^2 N}{2\sigma^2}. \tag{16}$$

The bound on $J(\mathbf{t})$ in Eq 16 can be converted into a bound on the noncentrality parameter from Eq 10 so that the content of the corollary can be expressed in units of power (Sect 5.2). We also note that the worst-case power of the fixed-period model provides a lower bound on the power of free-period rhythm detection after Bonferroni correction, as clarified further in S1 Text Sect S1-1.7. As a result, designs that are constructed to improve the worst-case fixed-period power should be understood as indirectly improving the multiple-test corrected free-period power.

Since the study investigates multiple rhythms, we assume that it has a higher sample size than the studies considered in the previous section. The brute force searches in Sect 3.1 are not tractable at high sample size (S5 Fig), so we developed an alternative optimization approach, stated formally in Theorem 3.3 and derived in Sect 5.2.2.

**Theorem 3.3** (Mixed-integer conic program for power optimization). *Worst-case power maximization with discrete time*

$$\text{maximize} \quad \min_{f \in \mathcal{F}} \xi_{\min} \left( B(\mu; f, \tau)^{-1} \right), \tag{17}$$

$$\text{subject to} \quad \mu \in \{0, 1\}^d, \quad \sum_{i=1}^{d} \mu_i = N, \tag{18}$$

*is equivalent to the following mixed-integer conic programming problem*

$$\text{maximize} \quad \eta, \tag{19}$$

$$\text{subject to} \quad X(f; \tau)^T \text{diag}(\mu) X(f; \tau) - \eta I \geq 0, \quad f \in \mathcal{F}, \tag{20}$$

$$\eta \in [0, N/2], \quad \mu \in \{0, 1\}^d, \quad \sum_{i=1}^{d} \mu_i = N, \tag{21}$$

*in which $X(f, \tau)$ is the design matrix of the one-frequency cosinor model evaluated at frequency $f$ on the partition $\tau$, and $B(\mu; f, \tau)$ is given by*

$$B(\mu; f, \tau) = H \left( X(f, \tau)^T \text{diag}(\mu) X(f, \tau) \right)^{-1} H^T, \quad H = \begin{bmatrix} 0 & 1 & 0 \\ 0 & 0 & 1 \end{bmatrix}. \tag{22}$$

We illustrate the applicability of Theorem 3.3 to bifrequency design optimization, the simplest nontrivial example of discrete period uncertainty. In this setting, the study investigates a pair of frequencies $(f_1, f_2)$ and aims to maximize the worst-case power at both frequencies for a given sample size. The widest frequency separation corresponded to resolving a pair of periods $T = 24hr$ and $T = 2hr$ with a sample size of $N = 12$ measurements. Interestingly, the optimal solution contained repeating patterns in its measurement schedule (Figs 2A and S6A). While the equispaced design suffered from

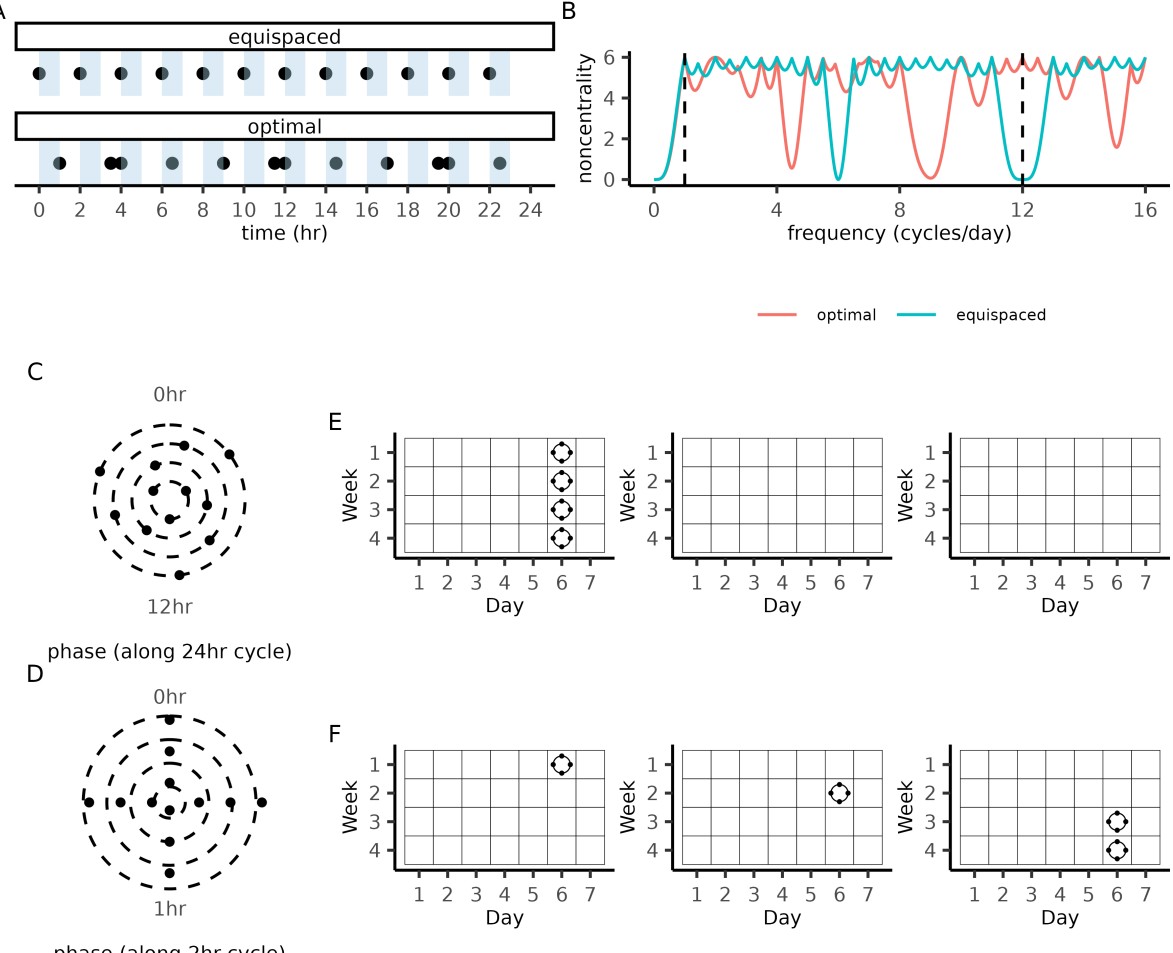

**Fig 2. Globally optimal designs for discrete period uncertainty. (A)** Measurement collection times of an equispaced design (top) and a bifrequency optimal design (bottom) for detecting 2hr and 24hr rhythms. **(B)** Power (y-axis) as a function of frequency (x-axis) for the equispaced and bifrequency optimal design. The two frequencies included in the optimization problem are indicated by the dashed vertical lines. **(C)** Bifrequency optimal design plotted as phases of a 24hr cycle. The values of the radial coordinate were chosen to emphasize that the design can be split into four equiphase designs, each containing three measurement times. **(D)** Bifrequency optimal design plotted as phases of the 2hr cycle. The radial position of each point represents its phase along the 24hr cycle. **(E)** Days marked with circles indicate that $n = 4$ equispaced circadian measurements are to be collected. Repeating this schedule four times over the course of a year, produces a trifrequency optimal design with measurements equispaced along circadian (24 hr), circalunar (28 day), and circannual (12 × 28 = 336 day) cycles. **(F)** An alternative measurement schedule that distributes the measurements across all three months while maintaining the equiphase property.

a low noncentrality parameter at the 2hr rhythm (Fig 2B), the optimal design reached its theoretical maximum value (Theorem 3.2) at both frequencies of interest. This ability to resolve both frequencies without sacrificing power at either frequency was not observed in 10,000 randomly generated designs (Fig S7A) or when additional frequencies were included in the design problem (Fig S7B).

Closer examination of the bifrequency optimal designs revealed a simple explanation for their ideal power balance; they can be split into equispaced designs when visualized as phases of each period in the design problem (Fig 2C and 2D) and therefore inherit the optimality properties of equispaced designs (S1 Text Sect S1-1.5). Since optimal designs can be constructed by simply ensuring that they satisfy this equiphase property, we used this approach to construct optimal designs for detecting circadian, circalunar, and circannual rhythms in a 12 month experiment (Fig 2E and 2F; Fig S7C).

### 3.3 Continuous period uncertainty

**3.3.1 Optimization of free-period power.** In experiments where the period of rhythmic activity must be estimated across a continuous range (i.e. longer than an hour and shorter than a day), the closed-form expression for statistical power from Theorem 3.1 is not directly available. We rely instead on a permutation test based on the amplitude estimate of the free-period cosinor model (Sect 5.1). Estimating the power of the permutation test is computationally expensive, so we instead derive a mean-squared bound that can be evaluated more efficiently (S1 Text Sect S1-2). We use this bound to construct candidate designs and then benchmark their performance using the power of the full permutation test.

Optimization of the bound requires specification of the frequency window $[f_{min}, f_{max}]$ and the sample size of the study. Improvements relative to equispaced designs were only observed when the frequency window included the Nyquist rate of the equispaced design ($f_{Nyq} = N/2$) and when the signal amplitude was sufficiently high (Fig 3A). Although the permutation bound reduced the compute time compared with the full permutation test, it remained much slower than the objective

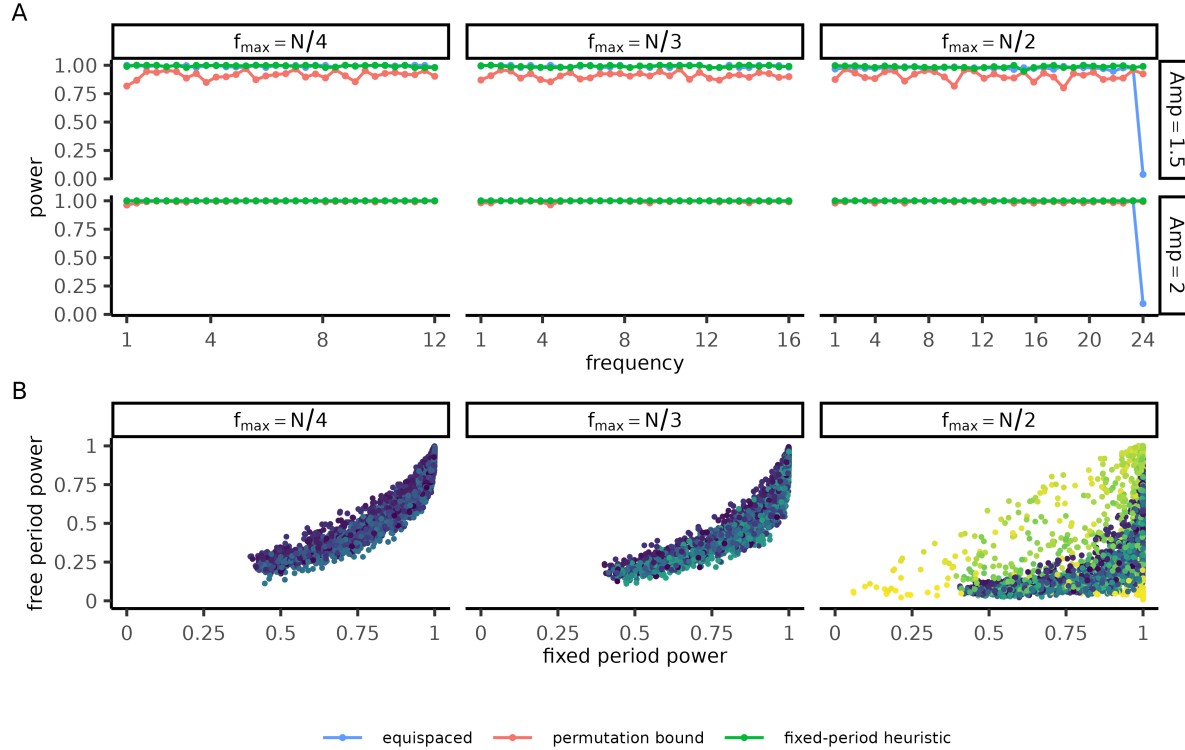

**Fig 3**. **Irregular sampling improves power near the Nyquist rate in free-period rhythm detection. (A)** Permutation tests were performed with the $T_\infty$ amplitude test-statistic (Sect 5.1). Independent signals were generated across all included frequencies and acrophases for each frequency window and amplitude (panels). Curves report the minimum power over acrophase (y-axis) as a function of frequency (x-axis) for equispaced and irregular designs (color), each with $N = 48$ measurements. Irregular designs were generated either by maximizing the permutation bound or using the worst-case fixed-period power as a heuristic. **(B)** Signals were simulated with $N = 12$ equispaced measurements spanning the slowest cycle in the frequency window $[1, f_{max}]$. Each point represents an estimate of the fixed-period (x-axis) and free-period (y-axis) cosinor power for a signal of random amplitude, acrophases, and frequency (color). Parameters: (A) each frequency window was discretized using $N_{freq} = 24$ and $N_{acro} = 8$ and $N_{samp} = 10^2$ independent samples of Gaussian white noise. Each of the $N_{freq} \times N_{acro} \times N_{samp}$ signals were then permuted $N_{perm} = 10^2$ for the permutation test. The $T_\infty$ test statistic was discretized using $N_{freq} = 48$ frequencies. For the equispaced design, $T_\infty$ becomes singular at the Nyquist rate so the frequency grid was restricted to only include frequencies $f \leq .99 f_{Nyq}$. The differential evolution design was generated using the same parameters as Fig 4. Parameters (B): $n = 3000$ signals were generated in each panel with amplitude $\sim \text{Unif}([1,3])$, acrophase $\sim \text{Unif}([0, 2\pi))$, frequency $\sim \text{Unif}([1, f_{max}])$. Power was estimated using $N_{perm} = 10^3$ and $N_{samp} = 500$ independent samples for each frequency. The $T_\infty$ test statistic was discretized using $N_{freq} = 500$ frequencies.

functions considered in earlier sections (S8 Fig). This limitation motivated us to ask whether comparable designs could be obtained using a simpler heuristic approach. To this end, we evaluated the worst-case fixed-period power as a candidate heuristic. Fixed-period and free-period power were strongly correlated across a broad range of signals (Fig 3B). When designs were optimized to maximize fixed-period power across the frequency window, they maintained high free-period power at all frequencies including the Nyquist rate (Fig 3A). Our results suggest that our heuristic approach – generating designs using fixed-period power and benchmarking them with free-period power – is an practical and effective way to optimize power under continuous period uncertainty.

### 3.3.2 Worst-case fixed-period power under continuous period uncertainty.

In this section we treat heuristic design optimization as a standalone problem and investigate the structure and robustness of candidate solutions. Designs were constructed to maximize the power heuristic on various frequency windows and sample sizes using PowerCHORD's differential evolution method. The greatest improvements to worst case for each sample size $N$ were observed when the Nyquist rate $f_{Nyq} = N/2$ was included in the frequency window (Fig 4A). Optimization of other windows produced at best a slight change in power. In comparison to equispaced designs, the power of irregular designs had much weaker power fluctuations around the Nyquist rate (Figs 4B and S3B). Away from the Nyquist rate, the equispaced and irregular designs exhibited power fluctuations that diminish in intensity as the sample size increases. Hence, for a large enough sample size, irregular designs improve worst-case power while maintaining homogeneous power levels throughout the frequency-acrophase cylinder.

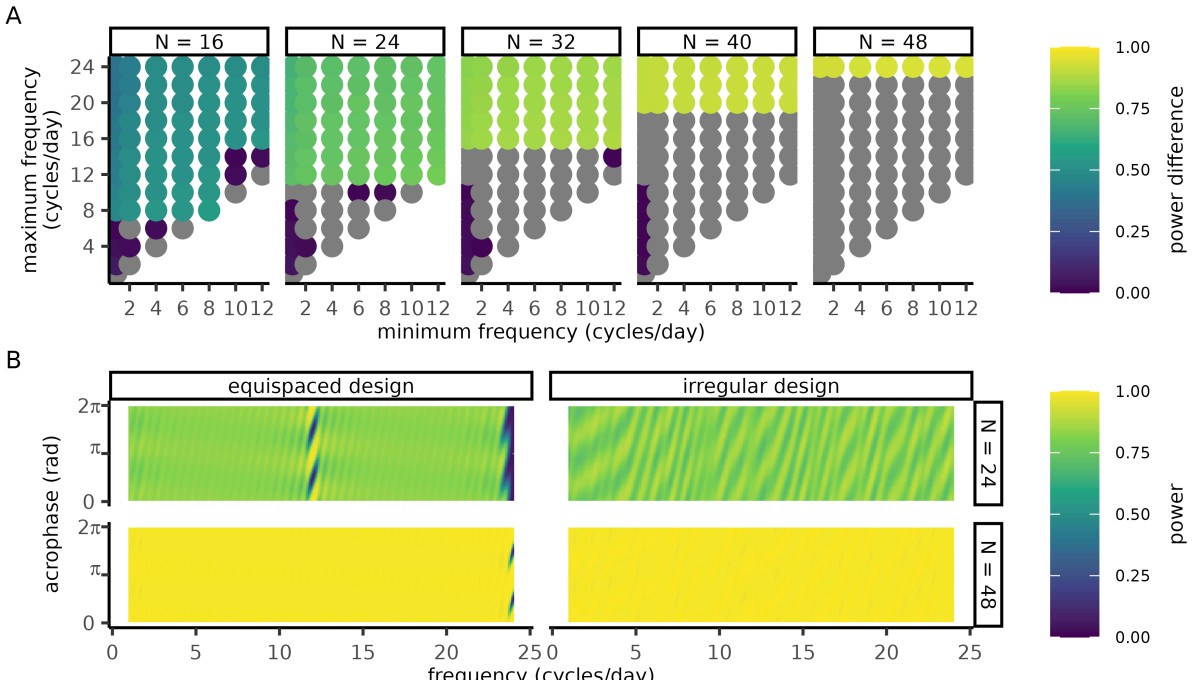

**Fig 4**. **Irregular sampling improves power in experiments with continuous period uncertainty.** Irregular designs were optimized by differential evolution to detect signals whose frequency is in a window $f_{min} \leq f \leq f_{max}$. **(A)** Differences in power between equispaced designs and optimized irregular designs where the $x$ and $y$ axes represent the upper ($f_{max}$) and lower ($f_{min}$) limits of each frequency range of interest. The color of each circle represents the difference in worst-case power between an irregular and equispaced design with the same number of measurements (panels). Grey circles correspond to negative power differentials. **(B)** Heatmap of power as a function of frequency ($x$-axis) and acrophase ($y$-axis) where color represents the power. **(A-B)** Simulated cosinor parameters: amplitude $A = 1$ and noise strength $\sigma = 1$. Each differential evolution was run with 1hr of compute time with parameters $CR = 5 \times 10^{-2}$, $N_{pop} = 10^3$, $\varepsilon = 5 \times 10^{-2}$ (see Sect 5.2.4). Details on power calculation at the Nyquist rate are given in S1 Text Sect S1-1.4.

Some of the measurements in irregular designs are spaced farther apart than in equispaced designs of the same sample size. These measurement gaps could make the design undesirably sensitive to the timing of specific measurements. To investigate this sensitivity, we generated "jittered" irregular and equispaced designs by adding Gaussian white noise to each measurement time (Fig 5A) and calculated the worst-case power of the increasingly perturbed designs. As the noise intensity increased, the power improvements of the irregular design declined as both designs converged in power as measurement times were effectively being sampled from the same distribution (Fig 5B). For a sample size of 24, the power difference between the irregular and equispaced designs reached half of its full difference when the measurement error was 9 minutes, while for a larger sample size of 48 this difference diminishes more quickly at a measurement error of 5 minutes. Irregular designs always outperformed the eventual random design ($\Delta = 0.18, 0.14, 0.04$ for $N = 24, 32, 48$, respectively). Since timing error is likely to be on the order of minutes, our power improvements are robust to errors that occur on a realistic timescale.

Our irregular designs were optimized for cosinor analysis, yet the data they produce may be used for other purposes such as spectral analysis. To determine the applicability of our irregular designs in such a context, we performed periodogram analysis on simulated datasets. We made use of the Lomb-Scargle periodogram, a common method developed for the study of irregularly sampled data [33,34], and applied a hypothesis test derived from this method [35]. We generated independent simulated datasets for a range of frequencies, each made up of white noise and oscillatory signals with uniformly random acrophases. We quantified performance using the area under an ROC (receiver operator characteristic) curve [36] because it can be calculated without knowing the false positive rate, and this rate could be affected by switching from cosinor to periodogram analysis. Relative to equispaced designs, irregular designs exhibited weaker fluctuations in their AUC (area under the curve) score at frequencies below the Nyquist rate (Fig 6A). At the Nyquist rate of the equispaced design ($f_{Nyq} = 20$), the irregular designs dramatically outperformed the equispaced design. Irregular designs performed well only at the frequencies included in their optimization (Fig 6B), emphasizing the importance of choosing a realistic frequency range before optimizing an experimental design. Although they were constructed for cosinor-based analysis, PowerCHORD designs remain applicable in a broader statistical context.

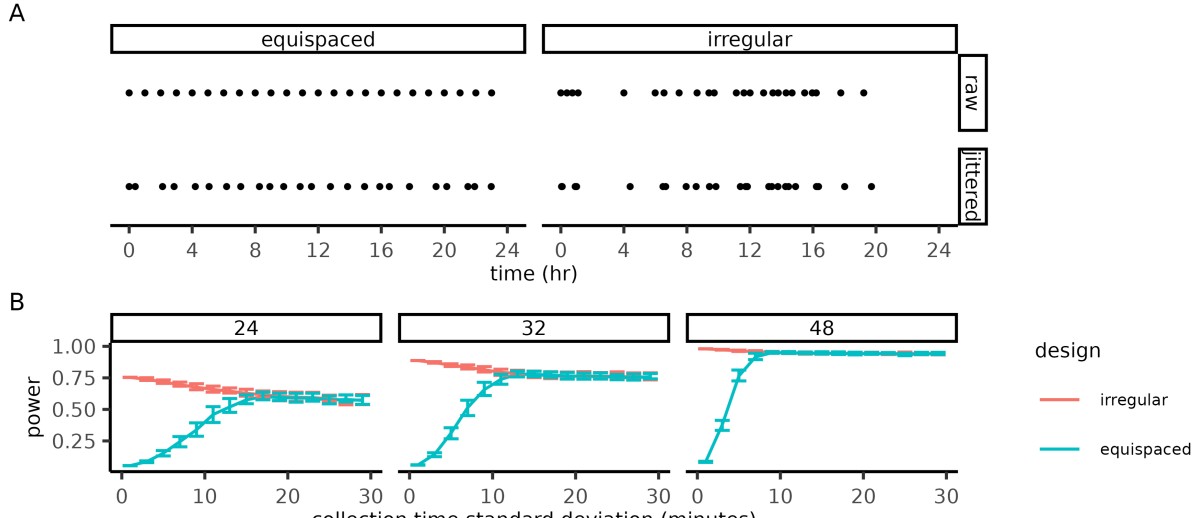

**Fig 5. Irregular designs are robust to perturbations in measurement timing. (A)** Measurement schedules before (top) and after (bottom) jittering. Jittered designs were generated by randomly perturbing the measurement times of equispaced and irregular designs with Gaussian white noise. **(B)** Worst-case power of jittered designs (*y*-axis) as a function of noise intensity (*x*-axis) for various sample sizes (panels). The bars indicate the interquartile range for ensembles ($n = 100$) jittered designs. Irregular designs were optimized for the frequency window $f_{min} = 1$ and $f_{max} = 24$ using differential evolution with same parameters as Fig 4. Power was calculated assuming an amplitude $A = 1$.

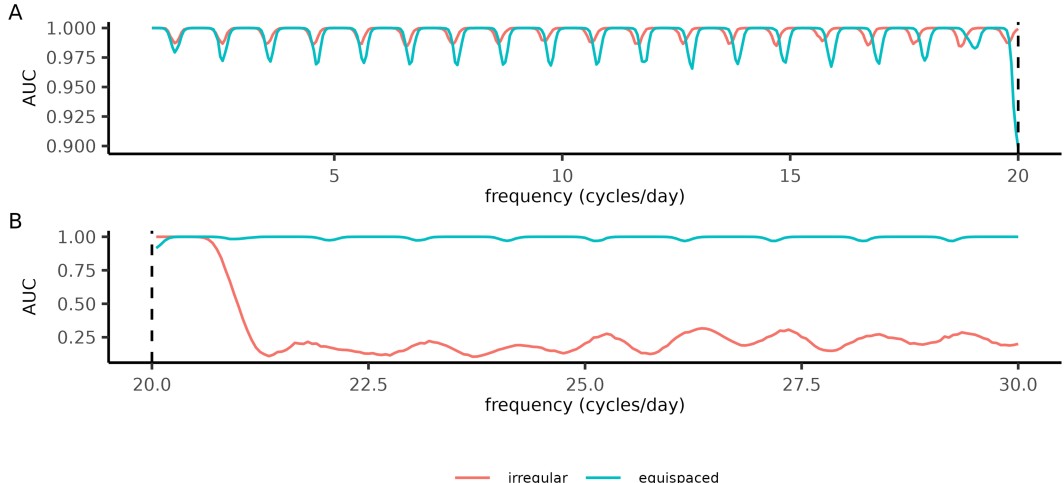

**Fig 6**. **Irregular designs improve simulated periodogram analysis at frequencies up to the Nyquist rate. (A)-(B)** Oscillations were detected using periodogram analysis with measurements ($N = 40$ samples) from either an equispaced or irregular design optimized for the frequencies $1 \leq f \leq 20$. Each signal in the dataset was assigned an oscillatory (amplitude $A = 2$, noise strength $\sigma = 1$) or non-oscillatory (amplitude $A = 0$, noise strength $\sigma = 1$) state with equal probability. The acrophase of the oscillatory signals was assigned uniformly at random ($\phi \sim \mathrm{Unif}([0, 2\pi))$). **(A)** Performance of the irregular and equispaced designs at detecting oscillations across frequencies (x-axis) included in the optimization, summarized by the AUC score (y-axis) of a receiver operator characteristic curve with p-values generated from a Lomb-Scargle periodogram. The AUC score for each frequency was computed by testing for oscillations in a dataset of oscillatory and white-noise signals ($n = 10^4$ signals per dataset). The dashed line indicates the Nyquist rate of the equispaced design. **(B)** The same analysis as (A) but at frequencies above the Nyquist rate of the equispaced design. Periodogram analysis was performed using the `lomb` library [37] and AUC scores were computed using the `pROC` library [38]. Irregular designs were generated using the same differential evolution parameters as in Fig 4.

## 4 Discussion

We presented a closed-form expression for the statistical power of cosinor-based rhythm detection which holds for arbitrary measurement times. This formula enables exploration of the rhythm detection design space in a manner that was infeasible with Monte-Carlo power estimation. It will be useful for regular statistical practice to answer questions such as the phase-detection bias introduced by the loss of a sample, or in datasets which may not have been optimally designed (e.g. a postmortem dataset [39]). Using our closed-form power expression, we proved that any given periodicity can be optimally detected by an equispaced design with $N > 3$ measurements per cycle. This optimality condition implies that certain groups of periods can be investigated together without any trade-off in power. In particular, the power will be phase-independent provided that the measurements are equally spaced when viewed as phases for each period under consideration. Cosinor analysis of equispaced designs continues to be performant when applied to non-sinusoidal waveforms as demonstrated in our simulations of non-cosinor rhythms and supplemental analysis of phase-invariant objective functions.

Application of optimization methods to various experimental scenarios improved the power relative to equispaced designs. For known period studies, PowerCHORD produced designs with a minimal loss in power relative to equispaced while removing the need for costly and logistically difficult overnight visits and avoiding sleep disruption biases. In the case of discrete period uncertainty, globally optimal designs were obtained under the simplifying assumption that measurements were confined to an underlying grid. The designs had noncentrality parameter $\lambda = N/2$ at all frequencies of interest and hence their performance was not limited by the restriction to an underlying grid. Examination of the bifrequency optimal designs revealed an "equiphase" property and led to simple optimal designs for simultaneous investigation of circadian, circalunar, and circannual rhythms. For continuous period uncertainty, we showed that a simple heuristic

based on fixed-period power was sufficient to improve free-period power relative to equispaced sampling near the Nyquist rate.

It may be possible to generate optimal designs in more challenging experimental contexts by improving the optimization methods in PowerCHORD. The brute-force searches could be accelerated to run at larger sample sizes using more efficient algorithms for parameterizing the design space [40] and related methods in optimal experimental design [41–44] could be adapted to our problem. While our analysis was focused on how measurement timing influences worst-case power, there are many closely related questions that could be considered in future work. First, timing optimization could be applied to adaptive study design. This application may be particularly important when signals are known to have greater phase velocity at particular phases of their cycle. Equipped with prior knowledge of the system, the prior would be updated using Bayesian inference and influence the optimization of measurement times for the next experiment [45, 46]. Second, the number of biological and technical replicates could be treated as decision variables in the optimization problem [14,47] to further improve power. Third, figures of merit relevant to the accuracy of parameter estimation or periodogram methods [35,37,48,49] could be prioritized in the study design. Beyond the scope of optimization, further improvements to rhythm detection can be achieved by higher quality data, innovative analytical methods and leveraging prior knowledge of the system [50].

In summary, we have demonstrated the benefits of integrating power optimization in the design of rhythm discovery experiments. Our theoretical results clarify both the advantages and limitations of equispaced designs and the conditions under which they are optimal. Our optimization methods are broadly applicable and can be tailored to specific experimental applications using our open-source PowerCHORD repository. Such studies will be more efficient than traditional approaches for exploring ranges of periods, and may expedite the discovery of novel biological rhythms.

## 5 Methods

### 5.1 Permutation testing with the free-period cosinor model

The free-period cosinor model is given by

$$y(t; f, \beta) = \beta_0 + \beta_1 \cos(2\pi f t) + \beta_2 \sin(2\pi f t) + \varepsilon(t), \quad \varepsilon \sim \mathcal{N}(0, \sigma^2) \tag{23}$$

with parameters $f, \beta_0, \beta_1, \beta_2 \in \mathbb{R}$. Assuming that the frequency lies within a window $f \in [f_{\min}, f_{\max}] \subset \mathbb{R}$, least squares regression with the free-period model reduces to linear regression, hence an optimal frequency $f^*$ can be written as

$$f^* \in \underset{f \in [f_{\min}, f_{\max}]}{\arg\min} ||\mathbf{y} - X_f \hat{\beta}_f||_2^2, \tag{24}$$

in which $\hat{\beta}_f = (X_f^T X_f)^{-1} X_f^T \mathbf{y}$ and $X_f$ is the cosinor design matrix evaluated at frequency $f$. Rhythm detection with the free-period model can be performed using a permutation test. In this framework, p-values are computed by determining how often the permuted data generates a value of a given test statistic that is more extreme than the observed value

$$\text{p} - \text{value} = \mathbb{P}_{\pi \in S_n} (T(\pi \mathbf{x}) > T(\mathbf{x})) = \frac{1}{n!} \sum_{\pi \in S_n} \chi (T(\pi \mathbf{x}) > T(\mathbf{x})), \tag{25}$$

in which $\chi$ denotes the indicator function

$$\chi(x) = \begin{cases} 1 & \text{if } x > 0, \\ 0 & \text{if } x \leq 0, \end{cases} \tag{26}$$

and $T(\mathbf{x})$ is a given test statistic. We analyze two test statistics based on the amplitude estimator of the free-period model

$$T_2(\mathbf{x}) = \int_{f_{\min}}^{f_{\max}} \hat{A}_f^2 \, df, \tag{27}$$

$$T_\infty(\mathbf{x}) = \max_{f \in [f_{\min}, f_{\max}]} \hat{A}_f(\mathbf{x}), \tag{28}$$

in which $\hat{A}_f = ||H\hat{\beta}_f||_2$ and $H$ is as given in Eq 11. Up to normalization, the $T_2$ test statistic corresponds to the mean squared amplitude across the frequencies of interest and $T_\infty$ corresponds to the largest amplitude. The latter quantity is likely to be more familiar to readers with experience in periodogram-based hypothesis testing. Both quantities are permutation invariant under the null distribution and are therefore amenable to permutation testing. While $T_\infty$ appears to be more sensitive to weak signals, $T_2$ is more analytically tractable. Consequently, we derive a lower bound based on the $T_2$ test statistic and use the $T_\infty$ test statistic to benchmark the performance of our designs.

### 5.2 PowerCHORD's optimization methods

PowerCHORD optimizes power using using a brute-force search, mixed-integer conic programming, a genetic algorithm, or differential evolution. Brute-force searches are computationally tractable if the sample size is low and the measurements are confined to an underlying grid (e.g. only sampling on the hour). Mixed integer conic programming is intended for higher sample size experiments with discrete period uncertainty, and the latter two methods are intended for optimization under continuous period uncertainty. Our methods are designed for studies with limited prior knowledge of the rhythm's parameters and therefore require minimal input data (sample size, frequencies under consideration, and timing constraints) for their optimization.

The optimization methods in PowerCHORD work directly with the eigenvalue problem from Eq 15 and their results can be converted back into power using the following identity

$$\lambda^* = \frac{A^2}{\sigma^2} \xi_{\min}\left(B(\mathbf{t};f)^{-1}\right), \tag{29}$$

in which $\xi_{\min}\left(B(\mathbf{t};f)^{-1}\right)$ is the minimal eigenvalue from Eq 12 and $\lambda^*$ is the lowest value of the noncentrality parameter across all acrophases for signals of amplitude $A$ and frequency $f$. Justification for Eq 29 is given in the proof of Lemma S1-1.12. In our analysis, we set $A = \sigma$ so that we may refer to the noncentrality parameter and minimum eigenvalue $\xi_{\min}\left(B(\mathbf{t};f)^{-1}\right)$ interchangeably. We also make use of the fact that the noncentral F distribution is a monotone function of its noncentrality parameter (see S1 Text Sect S1-1.2), which permits us to refer to the power of a design and its noncentrality parameter interchangeably.

**5.2.1 Brute-force search.** Suppose measurements are confined to a single cycle and can only be collected at times belonging to a fixed partition

$$\tau = \{k/N_t : k \in \{0, \ldots, N_t - 1\}\} \tag{30}$$

in which $N_t$ is the coarseness of the partition. Assuming that at most one measurement can be collected at each time in the partition, such scenarios are naturally represented by binary vectors $\mu \in \{0, 1\}^{N_t}$ which satisfy $\sum_i \mu_i = N$. Provided that $N$ and $N_t$ are not too large, it is feasible to search the entire space of binary vectors.

The search over binary vectors can be accelerated by making use of the rotational symmetry inherent to rhythm detection. Since any design $\mu \in \{0, 1\}^{N_t}$ will achieve equal performance to all of its cyclic translations, we need only consider equivalence classes of such binary vectors up to rotational symmetry. Following the convention from combinatorics, these

equivalence classes are referred to as fixed-density binary necklaces. For a partition of coarseness $N_t$ and sample size $N$, the number of equivalence classes of such necklaces is given by a sum over the factors of the greatest common divisor of the sample size and the partition coarseness

$$C(N, N_t) = \frac{1}{N_t} \sum_{j|\gcd(N,N_t)} \phi(j)\binom{N_t/j}{N/j}, \tag{31}$$

in which $\phi(\cdot)$ is Euler's totient function [51]. We recommend restricting $N_t \leq 72$ and $N \leq 10$ since this function grows rapidly with $N$ and $N_t$ (S5 Fig). An efficient algorithm for generating a representative from each equivalence class is given in [51]. We use their C implementation to generate the designs and search the design database to identify representatives of equivalence classes of optimal solutions.

There is an additional reflection symmetry present in the problem which could further improve the performance of the brute-force search. Notice that the time-reversal of any design will still have equivalent power, hence designs could be considered equivalent up to rotational and reflectional symmetry. In the terminology of combinatorics, these larger equivalence classes are known as "fixed-density binary bracelets". Efficient algorithms for generating representatives of the bracelets have been proposed [40] and would improve the performance of PowerCHORD if they were included.

**5.2.2 Mixed-integer conic programming.** Mixed-integer conic programming addresses discrete period uncertainty, meaning that multiple cycle lengths are considered and there is no longer a one-to-one mapping between linear time and phase (e.g. circular time vs linear time in [52]). While measurements are still encoded as binary vectors under the assumption of at most one measurement per linear time, multiple measurements may coincide when they are viewed as phases of a shorter cycle (e.g. timepoints $t = 3$hr and $t = 9$hr in a study that investigates 24hr and 6hr rhythms map to distinct phases of the 24hr cycle but coincide as phases of the 6hr cycle).

We provide a proof of Theorem 3.3 which justifies our use of mixed-integer conic programming. The main technical device in the proof is stated in the lemma below.

**Lemma 5.1** (Schur positivity, [53, Sect A.5.5]). *Let $Y \in \mathbb{R}^{n \times n}$ be a symmetric matrix partitioned as*

$$Y = \begin{bmatrix} A & B \\ B^T & C \end{bmatrix} \tag{32}$$

*with* $\det A \neq 0$ *and let* $S = C - BA^{-1}B^T$ *be the Schur complement of Y. Then we have* $Y \succ 0$ *if and only if* $A \succ 0$ *and* $S \succ 0$.

**Proof of Theorem 3.3.** Let $X = X(\tau; f)$ and $\tilde{X} = \tilde{X}(\tau; f)$. We seek a binary vector $\mu^*$ that satisfies

$$\mu^* = \underset{\mu \in \{0,1\}^d, \sum_i \mu_i = N}{\arg\max} \min_{f \in \mathcal{F}} \xi_{\min}\left(B(\mu; f, \tau)^{-1}\right), \tag{33}$$

in which $B(\mu; f, \tau)^{-1}$ can be expressed as

$$B(\mu; f, \tau)^{-1} = \tilde{X}^T \operatorname{diag}(\mu)\tilde{X} - \frac{1}{N}\mathbf{b}(\mu; f, \tau)\mathbf{b}(\mu; f, \tau)^T, \tag{34}$$

as justified by Lemma S1-1.13. Restricting frequency to a discrete set $\{f_1, \ldots, f_M\}$, we arrive at an optimization problem of the form

$$\max_{\eta \in \mathbb{R}, \mu \in \{0,1\}^d} \eta$$

$$\text{s.t.} \quad \sum_{i=1}^{d} \mu_i = N,$$

$$\tilde{X}^T \operatorname{diag}(\mu)\tilde{X} - \frac{1}{N}\mathbf{b}(\mu; f, \tau)\mathbf{b}(\mu; f, \tau)^T - \eta I \succeq 0 \quad \text{for } f \in \{f_1, \ldots, f_M\}. \tag{35}$$

The quadratic dependence on $\mu$ in Eq 35 can be reduced to a linear dependence by applying Lemma 5.1 with the matrix $Y$ given by

$$Y = \begin{bmatrix} N & \mathbf{b}(\mu; f, \tau)^T \\ \mathbf{b}(\mu; f, \tau) & \tilde{X}(\tau; f)\operatorname{diag}(\mu)\tilde{X}(\tau; f)^T \end{bmatrix} = X(\tau; f)^T \operatorname{diag}(\mu)X(\tau; f), \tag{36}$$

which gives

$$\max_{\eta \in \mathbb{R}, \mu \in \{0,1\}^d} \eta$$

$$\text{s.t.} \quad \sum_{i=1}^{d} \mu_i = N,$$

$$X(\tau; f)^T \operatorname{diag}(\mu)X(\tau; f) - \eta I \succeq 0 \quad \text{for } f \in \{f_1, \ldots, f_M\}. \tag{37}$$

$\square$

Optimal designs were generated by maximizing Eqs 19 and 21 using the CUTSDP method in YALMIP [54] together with Gurobi as a backend solver [55].

**5.2.3 Permutation bound optimization using genetic algorithm.** Fig 3 compares an equispaced design to a design constructed using the permutation bound from S1 Text Sect S1-2 as an objective function. To optimize the bound numerically, MATLAB's built-in genetic algorithm [56,57] was applied to the following constrained optimization problem

$$\max_{\mathbf{t} \in [0,1]^N} J_b(\mathbf{t}; f_{\min}, f_{\max}) \tag{38}$$

$$\text{s.t. } t_k + \varepsilon < t_{k+1}, \quad \text{for } 1 \leq k \leq N-1, \tag{39}$$

$$t_1 = 0, \quad t_{N/2+1} = \frac{1}{2}, \tag{40}$$

in which $J_b(\mathbf{t}; f_{\min}, f_{\max})$ is the permutation bound from Eq S163. The parameter $\varepsilon > 0$ ensures samples are not placed too close together and the equality constraints in Eq 40 ensure that samples remain spread out across the study and reduce the degrees of freedom in the problem.

**5.2.4 Differential evolution.** Differential evolution algorithms initialize a population of candidate solutions at random. Successive generations are constructed by taking component-wise linear combinations of the parents according to an algorithm-specific update rule. While many variations on this core idea have been studied [58], we chose to implement a relatively simple version of the algorithm.

Given an objective function $J : \mathbb{R}^N \to \mathbb{R}$, our implementation of differential evolution requires four hyper-parameters: the population size $N_{\text{pop}}$, the differential weight $\varepsilon \in [0, 1]$, the crossover probability $CR \in [0, 1]$, and the number of iterations

$N_{\text{iter}}$. At initialization, the population is represented by a matrix $x \in \mathbb{R}^{N_{\text{pop}} \times N}$, in which $N$ is the sample size, whose entries are independently and identically distributed as $x_{ij} \sim \text{unif}(0,1)$. To produce the next generation, the state of the $i$-th member of the population is updated by generating a random vector $u \sim \text{unif}(0,1)$, sampling three indices $a,b,c \in \{1, \ldots, N_{\text{pop}}\}$ without replacement to obtain

$$x_{ij} = \begin{cases} y_{ij} & \text{if } u \geq CR \\ x_{ia} + \varepsilon(x_{ib} - x_{ic}) & \text{if } u < CR. \end{cases} \tag{41}$$

If $J(y_i) < J(x_i)$, then the new state is accepted. The system repeats this process until a total of $N_{\text{iter}}$ generations have been produced. The highest scoring individual of the terminal population is then returned. A summary of the algorithm in pseudo-code is given below.

**Algorithm 1 Differential evolution in PowerCHORD.**

```
Require: N ≥ 3                                                          ▷ sample size
Require: N_pop ≥ 4,  N_iter > 0,  ε > 0,  CR ∈ (0,1)                     ▷ hyperparameters
Require: J : ℝ^N → ℝ                                                    ▷ objective function
  x_ij ← unif(0,1)
  while iter ≤ N_iter do                                               ▷ loop over generations
    for i ← 1 to N_pop do
      for j ← 1 to N do                                                ▷ generate candidate y_i
        u ← unif(0,1)
        y_ij ← x_ij
        if u < CR then
          {a,b,c} ⊂ {1,…,N_pop}                                       ▷ sample three distinct indices
          y_ij ← x_ia + ε(y_ib − y_ic)
        end if
      end for
      if J(y_i) < J(x_i) then                                         ▷ determine if y_i should replace x_i
        x_i ← y_i
      end if
    end for
    iter ← iter + 1
  end while
```

## Supporting information

**S1 Text. Proofs of mathematical results in the main text.**
(PDF)

**S1 Fig. An irregular design detects 24hr and 4hr rhythms at all acrophases.** Each oscillator ($n = 10^5$) in the simulated dataset was assigned a 24hr or 4hr period and a uniformly random acrophase $\phi \in [0, 2\pi)$. Measurements are simulated with Gaussian white noise at each measurement time. **(A)** Measurement schedules for (top) a traditional equispaced design, a (middle) fast-slow irregular design, and (bottom) a methodically constructed irregular design. The shaded bars represent 2hr increments and dots indicate sample collection ($N = 12$ samples for each design). **(B)** The average intensity of a Lomb-Scargle periodogram for each design. The true periods in the system are marked by the dashed vertical lines. **(C)** True acrophases of statistically significant oscillators ($p < 0.05$) detected by cosinor analysis at each of the true periods. Distributions with phase-dependent detection are shown in red to emphasize that the distribution's variability is due to a statistical artifact. Simulation parameters: amplitude $A = \sqrt{2}$, noise strength $\sigma = 1$, acrophase $\phi \sim \text{Unif}(0, 2\pi)$,

period $T = 4\text{hr}, 24\text{hr}$. The methodically-constructed design was generated using mixed-integer conic programming in PowerCHORD.
(TIFF)

**S2 Fig. The general power formula is necessary for accurate power analysis. (A)** Comparison of our power formula and the equispaced formula to Monte Carlo estimates for randomly generated designs in which $t \sim \text{unif}([0,1])$ for each measurement time $t$. On average, the equispaced formula tended to over-estimate the power of such designs. Parameters: sample size $N = 8$, amplitude $A = 2$, frequency $f = 1$, acrophase $\phi = \pi$, noise strength $\sigma = 1$. **(B)** We computed the power of designs $\mathbf{t}_{N,\kappa} = \kappa \mathbf{t}_N$, where $\mathbf{t}_N$ is an $N$ measurement equispaced design and $\kappa > 0$ is a scale factor. As $\kappa$ shrinks, the irregularity in the design becomes more pronounced and the equispaced formula diverges from the Monte Carlo power estimates. Parameters: $N = 24$, $A = 1$, $f = 1$, $\phi = 0$.
(TIFF)

**S3 Fig. Phase-dependence of equispaced power near critical frequencies. (A)** The power (color) of an equispaced design ($N = 24$ samples) evaluated at each frequency (x-axis) and acrophase (y-axis). The power is independent of phase when the frequency reaches $f = 1$ because the design is equiphase at this frequency. **(B)** The worst-case power of equispaced designs (sample size $8 \le N \le 40$) as a function of frequency, with frequency scaled relative to Nyquist rate of each design ($f_{\text{rel}} = f/f_{\text{Nyq}}$). Parameters: amplitude $A = 1$, noise strength $\sigma = 1$.
(TIFF)

**S4 Fig. Robustness of equispaced designs to cosinor assumptions.** Equispaced designs were compared to randomly generated designs ($n = 100$) of the same sample size ($N_{\text{meas}} = 12$) for detecting rhythms that fail to satisfy the assumptions of the cosinor model. **(A)** Each panel represents a type of signal: (1) standard cosinor, (2) cosinor with amplitude modulation, (3) cosinor with rhythmic noise of the form $\varepsilon(t) = \varepsilon(t)(1 + A\cos(2\pi f t - \phi))$ where $\varepsilon(t) \sim \mathcal{N}(0,1)$, (4) square wave, and (5) burst-like square wave. For all signal types, the amplitude $A$ and acrophase $\phi$ vary across simulations, and $\varepsilon(t) \sim \mathcal{N}(0,1)$ represents independent Gaussian noise. **(B)** True positive rate (y-axis) as a function of false positive rate (x-axis) for each type of signal (panels) and design (color). For each design and signal type, the curves were generated by simulating an ensemble of 5000 white noise signals (null model) and 5000 rhythmic signals (alternative model). Random designs were generated by uniformly random measurement times ($t \sim \text{Unif}([0,1])$). Parameters: sample size $N = 12$, amplitude $A \sim \text{Unif}([1,3])$, noise strength $\sigma = 1$, acrophase $\phi \sim \text{Unif}([0, 2\pi))$, frequency $f = 1$. Square-like waves were generated with random acrophase and duty cycle 0.5 for square waves and duty cycle 0.25 for burst-waves. For the rhythmic noise model, noise of rhythmic intensity (frequency $f = 1$) was included in simulations of both the null and alternative models.
(TIFF)

**S5 Fig. The number of experimental design equivalence classes grows rapidly with sample size.** For a given sample size (x-axis) and grid spacing (color), the number of design equivalence classes (y-axis) can be calculated using Eq 31. Designs are in the same equivalence class if they can be transformed into one another by a cyclic shift (i.e. $t \to t + k/N_t \mod 1$, for some $1 \le k \le N_t$ assuming measurements are in the interval $[0,1]$ and confined to a grid of spacing $1/N_t$).
(TIFF)

**S6 Fig. Low bias in significant parameter estimates from bifrequency optimal designs at both intended frequencies.** We computed bifrequency optimal designs for frequency priors $\nu = (1, f)$ with $f \in \{2, 4, 6, 8, 10, 12\}$ and a sample size $N = 12$. **(A)** Repetitive patterns appear in the measurement times of the bifrequency optimal designs. **(B-C)** Comparison of true amplitude and acrophase values to their cosinor estimates after filtering for statistically significance. With the exception of signals at integer multiples of the Nyquist rate ($f_{\text{Nyq}} = 6$), equispaced and optimal designs performed

similarly. At the Nyquist multiples, the optimal designs exhibited much less bias than equispaced designs. A randomly generated design ($t \sim \text{unif}(0, 1/12)$) with measurements confined to a short timescale was included as a reference. The random design performed poorly at low frequencies and improved as the higher frequency approaches the scale on which its points are distributed.
(TIFF)

**S7 Fig. Bifrequency and trifrequency optimal designs at critical frequencies. (A)** Non-centrality parameters for the two periodicities of interest (f=1 and f=12) for the optimal design (red dot) an equispaced design (blue dot) and an ensemble ($n = 10^4$) of randomly generated designs. The theoretical maximum value of the non-centrality parameter ($\lambda = N/2$; Theorem 3.2) is indicated by dashed lines. **(B)** Designs were generated to maximize power at the first $N/2$ harmonics ($f \in (1, \ldots, N/2)$) for each sample size $N$ (x-axis). The performance of each design is summarized by the lowest value of its non-centrality parameter (y-axis) across all harmonics included in the optimization. Color indicates convergence of the conic program within 1hr of computation time. The optimal noncentrality parameter in a single frequency design for each sample size ($\lambda = N/2$) is shown for reference (dashed line). For sample sizes $1 \leq N < 12$ measurements were confined to a 36 point grid, for $12 \leq N < 24$ a 48 point grid, and for $24 < N$ a 96 point grid. **(C)** The trifrequency optimal design with all measurements confined to the first month achieves phase independent power at 24 hr (circadian), 28 day (circalunar), and $12 \times 28 = 336$ day (circannual) periods. Parameters: Amplitude $A = 1/\sqrt{2}$.
(TIFF)

**S8 Fig. Compute times for exact and approximate power methods** Compute time (*y*-axis) is shown as a function of **(A)** the number of permutations (*x*-axis) and **(B)** sample size (*x*-axis) for each power method (color). Parameters: sample size fixed at $N = 24$ in (A) and permutations fixed at $N_{\text{perm}} = 10^3$ in (B). For both panels, noise samples $N_{\text{samp}} = 10^3$, frequency $f \sim \text{Unif}([0, 1])$, acrophase $\phi \sim \text{Unif}([0, 2\pi))$, amplitude $A \sim \text{Unif}([0, 1])$, $\mathbf{t} \sim \text{Unif}([0, 1]^N)$, and the $T_\infty$ test statistic was discretized using $N_{\text{f}} = 10^3$ frequencies.
(TIFF)

# Acknowledgments

We give thanks to Michael Tackenberg and John Limanto for providing thorough and insightful feedback on this manuscript.

# Author contributions

**Conceptualization:** Matthew Carlucci, Arturas Petronis, Adam Stinchcombe.

**Formal analysis:** Turner Silverthorne, Matthew Carlucci.

**Funding acquisition:** Arturas Petronis, Adam Stinchcombe.

**Investigation:** Turner Silverthorne.

**Software:** Turner Silverthorne, Matthew Carlucci.

**Supervision:** Adam Stinchcombe.

**Visualization:** Turner Silverthorne.

**Writing – original draft:** Turner Silverthorne, Matthew Carlucci, Adam Stinchcombe.

**Writing – review & editing:** Turner Silverthorne, Matthew Carlucci, Arturas Petronis, Adam Stinchcombe.

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
