## [Decision Letter · Decision Letter 0]

27 Jul 2025

PCOMPBIOL-D-25-00228

PowerCHORD: constructing optimal experimental designs for biological rhythm discovery

PLOS Computational Biology

Dear Dr. Stinchcombe,

Thank you for submitting your manuscript to PLOS Computational Biology. After careful consideration, we feel that it has merit but does not fully meet PLOS Computational Biology's publication criteria as it currently stands. Therefore, we invite you to submit a revised version of the manuscript that addresses the points raised during the review process. Both reviewers find your work interesting, but both raise questions about your use of the cosinor model, which you need to address.

Please submit your revised manuscript within 60 days Sep 26 2025 11:59PM. If you will need more time than this to complete your revisions, please reply to this message or contact the journal office at ploscompbiol@plos.org. Please include the following items when submitting your revised manuscript:

We look forward to receiving your revised manuscript.

Kind regards,

Marc Robinson-Rechavi

Academic Editor

PLOS Computational Biology

Ilya Ioshikhes

Section Editor

PLOS Computational Biology

**Journal Requirements:**

4) We notice that your supplementary Figures are included in the manuscript file. Please remove them and upload them with the file type 'Supporting Information'. Please ensure that each Supporting Information file has a legend listed in the manuscript after the references list.

5) We note that your Data Availability Statement is currently as follows: "All relevant data are within the manuscript and its Supporting Information files.". Please confirm at this time whether or not your submission contains all raw data required to replicate the results of your study. Authors must share the “minimal data set” for their submission. PLOS defines the minimal data set to consist of the data required to replicate all study findings reported in the article, as well as related metadata and methods (https://journals.plos.org/plosone/s/data-availability#loc-minimal-data-set-definition).

**Reviewers' comments:**

Reviewer's Responses to Questions

**Comments to the Authors:**

Reviewer #1: This manuscript presents PowerCHORD, which aims to aid design of studies interested in detecting rhythms of multiple possibly frequencies by choosing optimal timing of the sample collection times.

The authors point out that the most common study design (equidistant spacing) has short comings when there are multiple frequencies of interest.

Moreover, many other designs have biased acrophase estimation.

This is, to my knowledge, a novel approach to an overlooked problem and is of significance given its potential applications.

I particularly like the practical example of time-constrained designs where samples cannot be taken during sleep periods as well as the consideration of inexact measurement timing.

The paper is well-written and interesting throughout.

I have successfully run their provided R implementation and verified most of their proofs.

However, I do have one major concern that needs to be addressed in addition to some smaller points.

Major points

------------

1. The application proposed here is to design experiments where the period is unknow or partially unknown.

When analyzing the data from such an experiment, one must therefore use a cosinor model with an unknown period.

However, the power formula given in Theorem 3.1 is for a cosinor model with a fixed and known period.

The authors minimize this known-period cosinor power over all periods of interest, but this is not the same as computing the power of the unknown-period cosinor model.

The unknown-period model is a non-linear model (at least when the space of periods of interest includes an interval, as is done at points in this manuscript) and has complications such as aliasing (there is no longer a unique least squares or maximum likelihood solution in some cases).

It is possible (though I have not verified either way) that these results are therefore only approximations of the true power - and if so, the accuracy of this approximation must be assessed through simulation, etc.. The periodogram Lomb-Scargle analysis is already a step in this direction, but the implications or limitations involved must be addressed explicitly.

Minor points

------------

2. Top of page 3: Citations 20,21 are for two text books - can the authors be more specific about what is meant? Is there a specific page rage? Are these for definition of "optimal performance" or do they provide a proof of the claim that equidistant spacing is optimal for fixed-period cosinor analyses? It is unclear to the reader.

3. Is the direction of inequality in equation (16) correct? I would have expected the chosen lambda to out perform the one in equations 12-14 instead of underperform. Why not just use equation 12 instead then?

4. Section 5.2 Brute-force: It's not obvious to my why we can represent these as binary vectors - i.e., why is it never the case that two or more samples could be taken at the same time? Clearly in some applications, that wouldn't be done for practical reasons, but for others, it's quite common to take independent samples at the same time point

5. Lemma S1.4: V_ell is not a subspace (it does not contain 0, for example). It looks like you can just redefine V_ell to be the subspace of V_u that is orthogonal to V_r.

For the second equality, it's worth noting that the Q(Q^TQ)^{-1}Q^T is easily verified to be an orthogonal projection matrix (P^2 = P and P = P^T) from the definition of Q. Without that, computing the null space is not sufficient.

6. Lemma S1.7: Why are P_A and P_B commuting? It appears to need another assumption.

7. maximziation -> maximization (section S1.2)

8. Figure S5: No (C) in figure caption? Hard to decipher these plots: I take it that the pairs on the right hand side ("{1,2}") are the frequency priors and the fmin=1 and fmax = the second number of this? Also, please spell out for the reader what the Nyquist frequency is here (and elsewhere in the paper as applicable).

N = 12, so the Nyquist frequency is 6 and the integer multiple 12 is also affected? Worth saying for the reader as I, at least, always have to think through periods versus frequencies, and the sample size (and hence Nyquist frequency) is not constant throughout this paper.

Why don't all these subfigures (of B and C) have the same numbers of points drawn? Is there some filtering for a significant p-value or something? If so, I find that questionable when considering the biasedness of amplitude estimates, in particular.

Moreover, amplitude estimates appear biased upwards, especially at low amplitude. I guess that this is inherent to the cosinor model and amplitude being positive, but with the discussion of bias, this needs a comment and probably a clarification that bias is being used to refer to acrophase estimates only.

9. Monte Carlo is dismissed as infeasible for these study designs, presumably due to computational limits. Perhaps it would be possible to compare computational costs to PowerCHORD?

Reviewer #2: Uploaded as an attachment.

**Have the authors made all data and (if applicable) computational code underlying the findings in their manuscript fully available?**

Reviewer #1: Yes

Reviewer #2: Yes

PLOS authors have the option to publish the peer review history of their article (what does this mean?). If published, this will include your full peer review and any attached files.

Reviewer #1: **Yes: **Thomas G Brooks

Reviewer #2: No

**Figure resubmission:**
---

## [Decision Letter · Decision Letter 1]

14 Oct 2025

PCOMPBIOL-D-25-00228R1

Optimization of experimental designs for biological rhythm discovery

PLOS Computational Biology

Dear Dr. Stinchcombe,

Thank you for submitting your manuscript to PLOS Computational Biology. You will see that both reviewers' commended your revision. Reviewer 2 still has some suggestions. These are optional, but I am sending you the manuscript for revision to give you the opportunity to take these into account. Therefore, we invite you to submit a revised version of the manuscript that addresses the points raised during the review process.

Please submit your revised manuscript within 30 days Dec 14 2025 11:59PM. If you will need more time than this to complete your revisions, please reply to this message or contact the journal office at ploscompbiol@plos.org. Please include the following items when submitting your revised manuscript:

We look forward to receiving your revised manuscript.

Kind regards,

Marc Robinson-Rechavi

Academic Editor

PLOS Computational Biology

Ilya Ioshikhes

Section Editor

PLOS Computational Biology

**Reviewers' comments:**

Reviewer's Responses to Questions

**Comments to the Authors:**

Reviewer #1: The manuscript has received significant revisions and improvements. All of my points have been addressed.

My only new comment is that many figures come across as low-resolution and with significant compression artefacts in the version I am reviewing and I would advise the authors to make sure higher quality figures are available at publication.

Reviewer #2: The authors have done a commendable job at responding to most of my concerns. However, there is still one point which would benefit from a better discussion, and I urge the authors to address this before publication.

(1) The justification for use of a cosine model in Line 103 of the updated manuscript is fair, but not very informative. While Supp Fig 4 explores non-sinusoidal waveforms, this is not really the core issue in real datasets. Rather, the pervasive non-stationarity of the datasets is more important, for example, large variations in consecutive peak to peak distances in a time series dataset. While I understand that analysing non-stationary rhythms may not be within the scope of this manuscript, the authors should at least provide a more detailed discussion of this point such that readers are informed about the various assumptions and limitations involved with cosinor models, which are inherently stationary.

(2) Related to the point above, the authors respond to the question of non-stationarity by saying that "Power optimization is considerably more challenging for models where closed-form expressions

for the null and alternative distributions are unavailable." This seems to suggest that current methods for analysing non-stationary rhythmic datasets do not provide closed form expressions for the null and alternative distributions. This is however not true -- Gaussian Processes provide closed form solutions for both distributions, since both are multi-variate normals but with appropriate covariance matrices (see for example PMID 37769241; ref 24 in the author's updated manuscript). The authors might want to comment on this point for clarity.

**Have the authors made all data and (if applicable) computational code underlying the findings in their manuscript fully available?**

Reviewer #1: Yes

Reviewer #2: Yes

PLOS authors have the option to publish the peer review history of their article (what does this mean?). If published, this will include your full peer review and any attached files.

Reviewer #1: **Yes: **Thomas G. Brooks

Reviewer #2: No

**Figure resubmission:**
---

## [Editor Report · Decision Letter 2]

27 Oct 2025

Dear Dr. Stinchcombe,

We are pleased to inform you that your manuscript 'Optimization of experimental designs for biological rhythm discovery' has been provisionally accepted for publication in PLOS Computational Biology.

Best regards,

Marc Robinson-Rechavi

Academic Editor

PLOS Computational Biology

Ilya Ioshikhes

Section Editor

PLOS Computational Biology

---

## [Editor Report · Acceptance letter]

PCOMPBIOL-D-25-00228R2

Optimization of experimental designs for biological rhythm discovery

Dear Dr Stinchcombe,

I am pleased to inform you that your manuscript has been formally accepted for publication in PLOS Computational Biology. Your manuscript is now with our production department and you will be notified of the publication date in due course.

With kind regards,

Zsofia Freund
